# Estimation of Grassland Carrying Capacity by Applying High Spatiotemporal Remote Sensing Techniques in Zhenglan Banner, Inner Mongolia, China

**Pengyao Qin [1,2], Bin Sun [1,2], Zengyuan Li [1,2], Zhihai Gao [1,2,*], Yifu Li [1,2], Ziyu Yan [1,2] and Ting Gao [1,2]**

[1] Institute of Forest Resource Information Techniques, Chinese Academy of Forestry, Beijing 100091, China; qpyhebei@126.com (P.Q.); sunbin@ifrit.ac.cn (B.S.); lizy@caf.ac.cn (Z.L.); li.yifu1314@163.com (Y.L.); yanziyu96@163.com (Z.Y.); gaoting2167@126.com (T.G.)

[2] Key Laboratory of Forestry Remote Sensing and Information System, National Forestry and Grassland Administration, Beijing 100091, China

\* Correspondence: zhgao@ifrit.ac.cn

**Abstract:** Overgrazing directly leads to grassland degradation, which is a serious constraint to the sustainable development of animal husbandry. In drylands, grassland biomass is highly heterogeneous in space and time. It is difficult to achieve sustainable utilization of grassland resources by focusing only on the average annual carrying capacity assessment obtained from grass yield. Here, we proposed a novel approach for assessing grassland carrying capacity, taking Zhenglan Banner (County) in Inner Mongolia as the study area. First, monthly grass yield at 30 m spatial resolution was estimated, derived from Carnegie–Ames–Stanford Approach (CASA) model and spatial and temporal adaptive reflectance fusion model (STARFM). Then, based on the degree of sand mobility and degradation condition of typical steppe, the utilization patterns for sandy land and typical steppe in different grazing seasons were developed separately to obtain available grass yield. Finally, the carrying capacity at the Gacha (Village)-scale was estimated and the current livestock carrying status was evaluated to facilitate the grassland refined management. In Zhenglan Banner, the carrying capacity was 237.46 thousand cattle-units in summer. The grassland resources are being overgrazed, with an overload rate of 19.32%. At Gacha-scale, the maximum reasonable stock density was ranged from 0.06 cattle-unit/ha to 0.42 cattle-unit/ha. Fifty-one Gachas exhibited livestock overload. This study is expected to provide technical support and scientific reference data for ecological conservation and grassland management in the study area, as well as in dryland pastoral areas of northern China.

**Keywords:** drylands; grass yield; sand mobility; degradation condition; utilization pattern

## 1. Introduction

Global grasslands cover 3500 million hectares, accounting for about 26% of the world's land area and about 70% of the agricultural land area [1]. Grasslands are one of the most important types of terrestrial ecosystems on Earth because of their ecological functions, such as wind protection, sand fixation, soil and water conservation, air purification, and biodiversity maintenance. Furthermore, grasslands have an important economic value as the basis for animal husbandry and the production of mutton, beef, dairy, and other products [2,3]. In China, natural grasslands are the largest terrestrial ecosystem, accounting for 41.7% of the country's total land area and play an important role in livestock production and ecological restoration [4–6]. Animal husbandry based on the natural grasslands has been operating in China for almost a thousand years. Grazing is the most common grassland use in China and has become the most dominant way of human interference that affects grassland community structure and function [7,8]. However, over the past century, little attention has been paid to the coordination of ecological protection and production functions in the grasslands [9]. Overgrazing not only causes ecological degradation

but also greatly restricts the sustainable and healthy development of grassland animal husbandry [10–14].

Directly adopting grazing prohibition to restore the grassland ecosystem will not be conducive to the sustainable development of grassland animal husbandry. Meanwhile, several studies have shown that light to moderate grazing intensity practices can improve grassland productivity under certain environmental conditions [15–21]. Carrying capacity, a concept introduced to regulate grazing density and avoid overgrazing by livestock [22], can be summarized as the "ecologically sustainable stocking rates, considering vegetation production, site ecology and animal requirement" [23] (the "ecologically sustainable" component means ecological health was considered). In grassland management, it is expressed in terms of the number of livestock and days that they can graze on a certain area of grassland. The carrying capacity is calculated by developing a grassland resource utilization strategy (proper use factors) and estimating grass yield available for grazing and the daily feed requirements of livestock. Hence, estimating grassland carrying capacity is premised on ecological restoration and can be an effective tool to control and maintain the sustainability of grassland utilization, livestock production, and grassland ecosystem services [24].

For carrying capacity estimation, remote sensing can be an attractive application to estimate grass yield and track many immediate changes in grassland vegetation [25], as demonstrated by several attempts to study such capacity [26–30]. Remote sensing technology can solve the shortcomings of traditional methods, which are time-consuming, labor-intensive, and limited by management-related spatial scale [29]. In addition to providing large area coverage, remote sensing products also provide a higher temporal frequency of acquisition compared to traditional field sampling over large areas. Grass yield is one of the most significant indicators to determine carrying capacity [31–33]. Methods for achieving grass yield estimation using remote sensing can be broadly grouped into statistical, physical, and simulation models [27,29,34,35]. Currently, the statistical model is still a widely used method for estimating grass yield [36–39]. The model was pioneered by Tucker et al. [40] and Tucker et al. [41], who predicted grass yield by investigating the empirical relationship between remote sensing variables and ground truth data for inversion (See also Liu et al. [42]; Jin et al. [43]). Although the model is simple and the calculation is easy to operate, this method is not widely used for carrying capacity assessment. The reason for this phenomenon is that this model is built according to a specific location or seasonality, its transferability to other areas is unknown, and it is susceptible to the influence by vegetation types and non-vegetation factors (soil background, atmospheric conditions, topography, and bidirectional reflectance signature of land surface) [44]. The temporal and spatial transferability of grass yield estimation approaches is of great important for reproducible applications and coverage of large areas [45]. The second approach is a physical model, which refers to the estimation of grass yield from remote sensing information with the help of the relationship between the bidirectional reflection and grass yield. Presently, the models applied to estimate grass yield are mainly radiative transfer models [34]. Quan et al. [46] used leaf area index and dry matter content to estimate a plateau grass yield in China based on the radiative transfer model PROSAILH. Punalekar et al. [47] combined proximal hyperspectral and Sentinel 2A with a radiative transfer model (PROSAIL) to estimate pasture yield in a dairy farming context. Compared to statistical models, the radiative transfer model-based approach provides higher robustness and reproducibility for estimating grass yield on a large scale without the need to collect field measurements [46,47]. However, there is a well-known ill-posed problem when inverting the radiative transfer models [48–50]. Moreover, the models are complex and computationally intensive, and difficulties are encountered in data acquisition and noise removal. The third approach is the simulation model (also known as the process model), which is to model grass yield based on the net primary production (NPP) estimated from remote sensing [51–54]. The model emphasizes the description of the various processes acting within the grassland ecosystem [55], and its estimation results are more reliable. In recent years,

a large number of simulation models have been established, such as global production efficiency model (GLO-PEM) [56,57], bio-geochemical (BIOME-BGC) model [58], simple diagnostic biosphere model (SDBM) [59], TURC model [60], C-Fix model [61,62], vegetation photosynthesis model (VPM) [63,64], Lund university light use efficiency (LULUE) model [65], Eddy covariance-light use efficiency (ECLUE) model [66,67], organizing carbon and hydrology in dynamic ecosystems (ORCHIDEE) model [68], CENTURY model [69], and DeNitrification-DeComposition (DNDC) model [70], etc. Among the many models, a large majority of the studies investigated grass yield based on satellite data using a light use efficiency (LUE) model [32,34,71], suggesting that the LUE models are very effective and the most promising research tools [72]. The Carnegie–Ames–Stanford Approach (CASA) LUE model was mostly used among the published studies [54,73–75].

However, not all the estimated grass yield can be used for livestock rearing. If the grass utilization rate is too high, it will reduce the plant regeneration ability and the species diversity, and it will also affect the vegetation productivity and the sustainable use of grassland. On the contrary, if the grass utilization rate is too low, it is not conducive to the full utilization of grassland resources, and it is difficult to obtain the benefits of livestock production [20]. Thus, the development of a grassland utilization strategy to determine available grass yield is important for the coordination of grassland ecological restoration and husbandry sustainable development [76]. Zhang et al. [77] determined grassland utilization strategies for both cold and warm seasons based on grassland vegetation types and grazing utilization patterns with reference to the industry standard NY/T 635–2002 and estimated livestock carrying capacity for the Three-River Headwaters Region. Kuang et al. [30] developed a year-round grassland utilization strategy for 24 farms and ranches of the Hulunbuir Agricultural Reclamation Group based on grassland utilization patterns and assessed the ecological carrying capacity and the overloading condition of grassland. Neudert et al. [78] developed a grassland use strategy for obtaining maximum available grass yield in the continuously grazed area of the Greater Caucasus and calculated the potential grazing rate in the region. Hunt et al. [32] took Wyoming as a study area to determine grassland use strategies based on vegetation community type and calculated stocking rates per acre of animal unit month. To better cope with the complex characteristics of grassland ecosystem dynamics and uncertainty, Yu et al. [79] used slope, distance from water sources, and soil erosion intensity as limiting factors to formulate grassland use strategies and adjust the theoretical livestock carrying capacity. De Leeuw et al. [80] introduced a slope factor to further limit the amount of available grass yield on a slope greater than 10% when developing a grassland use strategy, thereby preventing the occurrence of soil erosion on a steep slope. The existing grassland use strategy focuses on the maximum available grassland yield and less on determining the sustainable available grassland yield from the current state of the grassland ecosystem.

Zhenglan Banner, Inner Mongolia, China, is located in the typical semi-arid grassland zone and the core area of Beijing-Tianjin Sand Source Region, which is an important ecological barrier for North China. Grassland-based livestock production is the foundation of the economy over there. Currently, the irrational use of grassland resources has made the grassland ecosystem more vulnerable, resulting in a poor ecological environment and seriously affecting local socioeconomic development [81]. How to achieve a balance between grassland ecological restoration and livestock development is an urgent issue.

In this context, this study aimed at proposing a novel approach to develop grassland utilization patterns and assess carrying capacity based on high spatiotemporal remote sensing techniques. This study taking Zhenglan Banner as an example for preliminary planning is expected to provide technical support and scientific reference for large-scale grassland conservation and utilization in both the study area and the entire dryland region of northern China. Specifically, the objectives of this study were to:

- develop spatiotemporal utilization patterns of grassland resource to obtain available grass yield, especially propose proper use factors with explicit spatial and temporal

scales based on the current condition of grassland resources and seasonal changes in grass yield; and

- achieve an elaborate estimation of carrying capacity at Gacha (Village)-scale in the study area by taking the temporal and spatial advantage of the available grass yield estimation results.

This paper is structured as follows: Section 1 sets out existing methods for the two key parameters of grass yield and grassland utilization strategy in carrying capacity estimation; detailed introductions to the proposed carrying capacity estimation approach are presented in Section 2; the approach was implemented in Zhenglan Banner and carrying capacity estimation results are shown in Section 3; the improvement of the grass yield estimation model and performance of the grassland resource utilization patterns are discussed in Section 4; Section 5 concludes with a summary of the results and directions for future work.

## 2. Materials and Methods

### 2.1. Study Area

The Zhenglan Banner is located in the south of Xilin Gol League, Inner Mongolia, China, geographically between 41°56′–43°11′ N and 115°00′–116°42′ E and covering a total area of about 10,000 km² (Figure 1) [81]. The terrain decreases from southwest to northeast with altitude ranging from 1200 to 1600 m. The climate belongs to the arid continental monsoon climate in the middle temperate zone. The annual average temperature is 1.7 °C, and average annual precipitation and evaporation are 366.8 mm and 1936.2 mm, respectively [82].

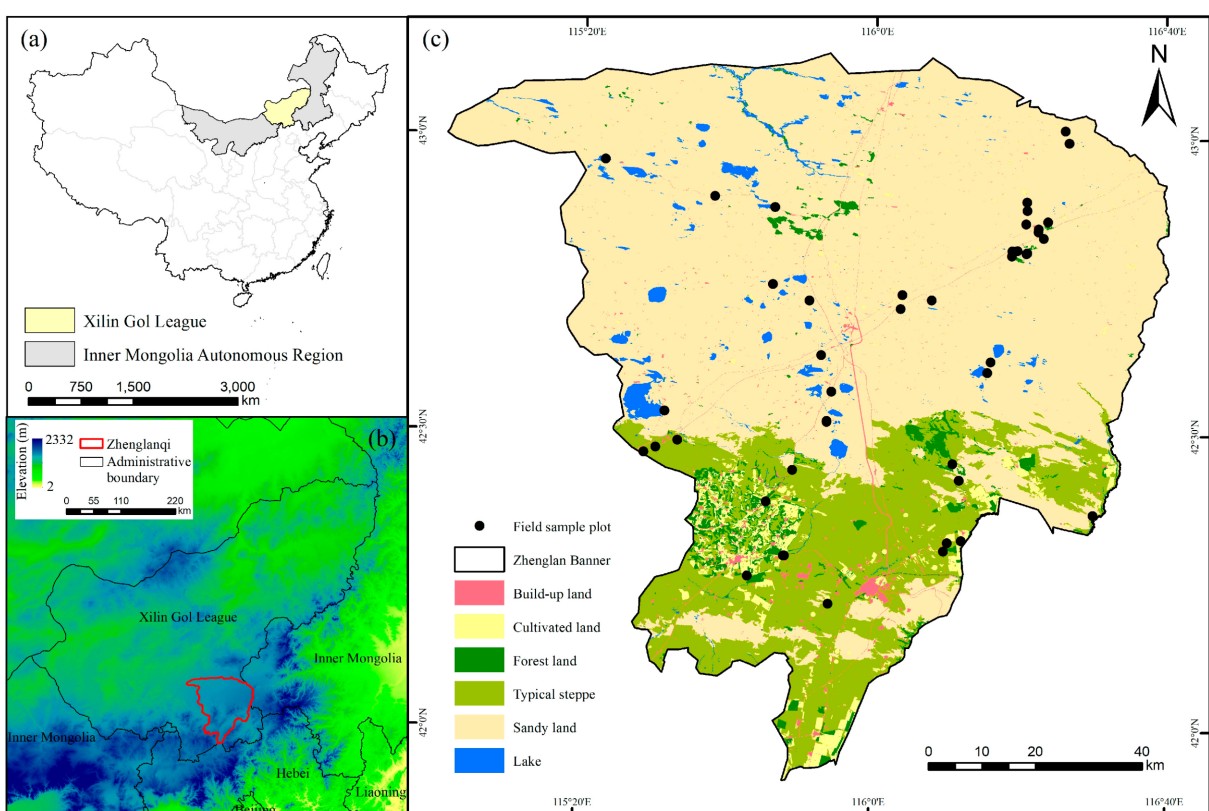

**Figure 1.** (**a**) Location of Xilin Gol League in China; (**b**) elevation distribution and location of the study area; and (**c**) land cover types in study area and the spatial distribution of grassland aboveground biomass field sample plots.

Natural grasslands are the advantageous resource in Zhenglan Banner, and the available grassland area accounts for 81.84% of the total land area. The natural grasslands in the northern part of the study area are distributed in the hinterland of the Otindag

sandy land, which is one of the four semi-arid sandy areas in China and is covered by shifting sandy land, semi-fixed sandy land and fixed sandy land [83]. The southern part grows in the typical steppe of temperate zone. Due to the spatial heterogeneity of ecological factors such as moisture, substrate conditions, and terrain, natural grasslands have formed different vegetation types, which can be roughly divided into sand vegetation and steppe vegetation [82]. Sand vegetation is the largest and representative vegetation type in Zhenglan Banner [84], including *Artemisia intramongolica* H.C.Fu., *Caragana microphylla* Lam., *Salix microstachya* Turcz. ex Trautv var. bordensis (Nakai) C.F.Fang, *Ulmus pumila* Linn., *Psammochloa villosa* (Trin.) Bor, *Polygonum divaricatum* Linn., *Corispermum mongolicum* Iljin, *Agriophyllum squarrosum* (Linn.) Moq., etc. Steppe vegetation is zonal vegetation in the area [84], such as *Leymus chinensis* (Trin.) Tzvel., *Stipa krylovii* Roshev., *Artemisia frigida* Willd., *Cleistogenes squarrosa* (Trin.) Keng, etc. This grassland vegetation usually turns green in early April and stops growing in early September [85].

Grassland animal husbandry is the leading industry and distinctive industry in Zhenglan Banner's economic development. In livestock production, herdsmen mainly adopt free-range grazing and supplementary feeding. The ecological status of Zhenglan Banner is extremely important because it is the closest typical grassland and desert land to Beijing and Tianjin. In recent years, a series of grassland conservation and construction national projects, such as the "Beijing-Tianjin sand source control", "returning grazing to grassland", "grassland ecological compensation and awarding", and "management of reclaimed grasslands in the agro-pastoralist zone", have been implemented in Zhenglan Banner [86–88]. The regional ecological environment has been greatly improved [89]. The local government has stepped up its efforts to restructure its animal husbandry, transformed how it grows, vigorously developed modern and efficient animal husbandry, and adopted a series of measures, such as "surround shift" and "raise more cattle and less sheep", to relieve pressure on grassland and improve economic efficiency [90]. However, the region is still affected by overgrazing with vegetation degradation. Therefore, ecological restoration and sustainable use of grasslands remain key issues [12,81].

*2.2. Data Collection and Processing*

2.2.1. Remote Sensing Data

In this study, 58 Landsat 8 Operational Land Imager (OLI) Collections Level-1 scenes (Path/Row: 124/30 and 124/31) during 2014–2016 and 9 Landsat 5 Thematic Mapper (TM) Collections Level-1 from June to August during 1985–1987 were collected from the U.S. Geological Survey (USGS) (http://glovis.usgs.gov/ (accessed on 11 March 2021)), which have a temporal resolution of 16 days and spatial resolution of 30 m. The Landsat data were preprocessed with radiance calibration, atmospheric correction, mosaicking, and clipping and were projected into Universal Transverse Mercator (UTM) 50 N with a World Geodetic System (WGS-84) datum.

However, it is difficult to ensure that high-quality cloud-free Landsat data images were available every month for calculation of monthly Normalized Difference Vegetation Index (NDVI). The MOD13Q1 NDVI data (NASA, https://ladsweb.modaps.eosdis.nasa.gov/ (accessed on 11 March 2021)) during 2014–2016 in the study area were obtained for the construction of high-resolution monthly NDVI datasets. The monthly MODIS NDVI data were merged by the maximum value composite (MVC) method. Furthermore, to eliminate abnormal changes caused by adverse effects, such as snow accumulation, cloud pollution, and data transmission errors, the Savitzky–Golay (S-G) filtering method [91] was used to reconstruct the monthly MODIS NDVI dataset. At the same time, all coarse-resolution MODIS NDVI data were resampled into 30 m spatial resolution and reprojected as UTM 50 N with a WGS-84 datum, consistent with Landsat data for further use in image fusion model.

2.2.2. Meteorological Data and Land Use and Land Cover (LULC) Data

The meteorological data and LULC data were used for monthly NPP estimation. Meteorological data (including monthly temperature, precipitation, and ground downward shortwave radiation) covering a period of 2014–2016 around the Zhenglan Banner, with a spatial resolution of 0.1°, were extracted from China meteorological forcing dataset (1979–2018) [92]. LULC data came from the land use change survey results of Zhenglan Banner in 2016, which were obtained by Technical Regulation for the Second National Land-use Investigation (TD/T 1014–2007) [93]. The LULC types were summarized into 5 categories including sandy land, typical steppe, farmland, build-up land, and water. All the above datasets, which were well quality controlled prior to delivery, were reprojected and resampled to the same spatial resolution as the monthly NDVI dataset. Besides, the meteorological data were resampled to the same temporal resolution as the monthly NDVI dataset.

2.2.3. Field Investigations

Considering that the official statistical data released may not be accurate, the livestock data based on Gacha as the statistical unit were derived from a walk-through survey to 109 Gachas in Zhenglan Banner in 2018, mainly including livestock number, species structure, age structure, and feeding methods. In the survey process, we divided into six groups and conducted questionnaires for each herding household in each Gacha to ensure the accuracy and comprehensiveness of the survey results.

The measured grass yield data were taken from 36 field sample plots obtained by the research group through the sample harvest method in July or August of each year from 2014 to 2016 in Zhenglan Banner grassland [81]. The data were used to validate the accuracy of grass yield estimation for the same location at the same time.

2.2.4. Other Data

The administrative boundary data of each Gacha in Zhenglan Banner were provided by the local government. The statistical auxiliary data such as livestock number and fenced grassland area were derived from the Zhenglan Banner Statistical Yearbook (2014–2016).

*2.3. Methodology*

2.3.1. Estimation of Grass Yield at High Spatial and Temporal Resolution

The estimation of monthly grass yield (GY, $g/m^2$) at 30 m spatial resolution during 2014 to 2016 was achieved based on the NPP prediction data with improved spatial resolution.

$$GY = \frac{cNPP}{0.47} \times fAG, \tag{1}$$

where cNPP is the cumulative value of NPP at a given period, which is expressed in $gC/m^2$, is converted to biomass using a biomass to carbon conversion factor of 0.47 C [94]. The fAG represents the proportion of above-ground growth to total growth, and the value taken in this study is 0.26 [94]. There is a premise here that grass yield can be obtained by accumulating monthly NPP. This is a reasonable assumption for grassland ecosystems where aboveground biomass dies off during the winter months. The performance of the predicting GY was evaluated by comparing the measured grass yield data with the GY predicted by this study's estimation model for the same year and sites. The terms of the accuracy were represented by Mean Absolute Error (MAE) and Root-Mean-Square Error (RMSE).

The estimation process for the NPP prediction data with improved spatial resolution was as follows. First, the spatial and temporal adaptive reflectance fusion model (STARFM), which was proposed by Gao et al. [95], in synergy between Landsat 8 OLI NDIV data with high spatial but low temporal resolution and MODIS NDVI data with low spatial but high temporal resolution, was applied for reconstructing a continuous and evenly distributed NDVI dataset with high spatiotemporal resolution (30 m and 1 month) during 2014 to 2016.

The core algorithm of STARFM was to determine pixel weights and conversion coefficients by identifying similar pixels between relatively high spatial resolution data and coarse spatial resolution data at the same time, following which determined relationship was used for the coarse spatial resolution data at another time to simulate the high spatial resolution data at that time. This model can greatly improve the accuracy of simulated NDVI data. Next, the monthly NPP was estimated using the CASA model. In this study, the two key parameters used in the CASA model, $\varepsilon$ and $\beta$, were optimized using both measured data and methods supported by our previous research [96,97] due to the sparse vegetation cover and relatively complex surface conditions in the study area. The parameters, therefore, fitted well to the characteristics of the vegetation and soil conditions. To some extent, the effects caused by the soil or sand matrix can be eliminated, resulting in relatively high accuracy estimates. Based on the simulated monthly NDVI data, synchronous meteorological data, and LULC data, the monthly NPP at 30 m spatial resolution was estimated during 2014 to 2016.

2.3.2. Development of Spatiotemporal Utilization Pattern for Grassland Resource Determination of the Degree of Sand Mobility and Steppe Degradation

Taking into account the characteristics of sandy land in Zhenglan Banner and the feasibility of implementing control measures, referring to the Technical Code of Practice on the Sandified Land Monitoring (GB/T 24255–2009) [98], fractional vegetation cover (FVC) was selected as a key indicator to grade the sand mobility. The mobility of sandy land was divided into five levels: shifting sandy land with FVC less than 10%, semi-shifting sandy land with FVC between 10% and 30%, semi-fixed sandy land with FVC between 30% and 50%, fixed sandy land with FVC between 50% and 70%, and full-fixed sandy land with FVC greater than 70%. Considering the influence of annual rainfall fluctuations on sand mobility, remote sensing images of three years from 2014 to 2016 were selected as data sources to reflect the general condition of surface cover in the 2010s.

Referring to the parameters for degradation, sandification, and salification of rangelands (GB19377–2003) [99] and combining with the degradation characteristics of typical steppe in Zhenglan Banner, the reduction rate of grass yield was selected as a key indicator for depicting the degradation state of typical steppe. By calculating the reduction rate of grass yield in the 2010s compared with that in the 1980s, the degree of steppe degradation was classified into four levels: severe degradation with a reduction rate greater than 50%, medium degradation with a reduction rate between 20% and 50%, light degradation with a reduction rate between 10% and 20%, and no degradation with a reduction rate less than 10%. To reflect the level of grass yield more accurately in different eras and reduce the influence of random factors such as annual precipitation on grass yield, the grass yield in the 2010s and 1980s was represented by averaging the 3-year grass yield from 2014–2016 and 1985–1987, respectively.

Development of the Spatiotemporal Utilization Patterns of Grassland

The grazing strategies for different grassland resource types were determined based on the degree of sand mobility and the characteristics of typical steppe (including the degree of degradation, current grass yield, and vegetation types). Then, based on the degree of sand mobility and typical steppe degradation, the proper use factors were obtained by referring to the calculation of rangeland carrying capacity (NY/T 635–2015) [100], experts interviews, and the livestock data survey results.

Based on the climate, vegetation growth rhythms, and grazing utilization characteristics of different seasons in the study area, the grazing seasons of grassland resource utilization were further divided to set different utilization intensity. In autumn and winter, the available grass yield mainly comes from the remaining grass yield of summer grassland resources after livestock gnawing and natural decline. To further improve the evaluation accuracy of the available grass yield in autumn and winter, the autumn and winter grass

preservation rate (80%) was introduced with reference to the Cui et al. [101] study on the dynamic of grass preservation rates in Xilingol grassland in the cold season.

### 2.3.3. Evaluation of Available Grass Yield

The available grass yield (AY, kg/ha) of the grassland resources was then evaluated using the equation:

$$AY = GY \times PU, \tag{2}$$

where GY is expressed in kg/ha, and PU is the proper use factors in spatiotemporal utilization patterns for grassland resources.

### 2.3.4. Assessment of Carrying Capacity and Current Livestock Carrying Status

The carrying capacity in number of Simmental cattle-unit/ha (CC) was calculated based on the available grass yield in different types of grassland resources during different grazing seasons:

$$CC = \frac{AY}{FI \times GD}, \tag{3}$$

where FI is the daily food intake of a Simmental cattle-unit ($kg \cdot day^{-1} \cdot$ one cattle-unit$^{-1}$) and GD is the number of grazing days (day). In China's pastoral areas, sheep unit is usually used as the standard livestock. All types of livestock were converted into standard sheep unit (conversion factors are detailed in NY/T 635–2015) [100], which is convenient for the government to count and control the number of livestock. An adult sheep weighing 45 kg and consuming 1.8 kg standard hay per day, or other livestock equivalent to this, is defined as one sheep unit (NY/T 635–2015) [100]. Based on the livestock data survey results in the study area and the local government's strategy of raising more cattle and less sheep [102], almost all the herders are mainly cattle breeders and the dominant breed of cattle is Simmental cattle. Zhenglan Banner has formed a grassland animal husbandry development model that focuses on feeding Simmental cattle [103]. It was more appropriate to choose the Simmental cattle-unit to calculate carrying capacity. Referring to NY/T 635–2015 [100], 1 Simmental cattle-unit equals 8 sheep-units and consumes 14.4 kg standard hay per day.

Under fully grazed conditions (without supplemental feeding), the livestock carrying status is calculated as

$$OR = \frac{CC - AS}{CC} \times 100\%, \tag{4}$$

where livestock overloading rate (OR, %) shows whether grazing activities have exceeded the carrying capacity of an area over a certain period of time. A positive result indicates that the grassland is under-exploited, and a negative result indicates that it is over-exploited. AS is the actual stocking rate in number of Simmental cattle-unit/ha.

## 3. Results and Analysis

### 3.1. Spatiotemporal Utilization Patterns for Grassland

### 3.1.1. Seasons Available for Grazing

The grazing time can be divided into three grazing seasons: (1) Summer (1 June to 30 September). This season is the peak season for the growth of vegetation, with the highest grass production, and livestock feeding method is mainly grazing. (2) Autumn and winter (1 October to 31 March). The grassland vegetation gradually dies off during this season, and the available grassland resource is dominated by the grasses remaining in summer. Livestock feeding method must be supplemented with forage other than grazing. (3) Spring (1 April to 31 May). This is the season of grassland vegetation emergence; overgrazing can seriously affect the growth of vegetation throughout the year. Livestock feeding method is to restrict grazing or even prohibit grazing.

### 3.1.2. Spatiotemporal Utilization Patterns in Sandy Land

The grazing utilization strategies and proper use factors of different sandy lands are shown in Table 1. Full-fixed and fixed sandy land with high vegetation coverage is the dominant grazing utilization type, but livestock-rearing control and spring banning grazing are needed to prevent enhanced sand mobility. Semi-fixed sandy land also has relatively high vegetation cover; spring banning grazing and livestock-rearing control are recommended. Shifting and semi-shifting sandy land has low vegetation cover and large areas of bare sand that are unsuitable for grazing.

**Table 1.** Spatiotemporal utilization patterns in different sandy lands.

| Sand Mobility | Grazing Strategies | Proper Use Factors | | | |
|---|---|---|---|---|---|
| | | Summer | Autumn and Winter | Spring | Annual |
| Shifting | Banning grazing all year round [1] | 0 | 0 | 0 | 0 |
| Semi-shifting | Banning grazing all year round [1] | 0 | 0 | 0 | 0 |
| Semi-fixed | Spring banning grazing [2] + Livestock-rearing control [3] | 30% | 80% × 30% | 0 | 47% |
| Fixed | Spring banning grazing [2] + Livestock-rearing control [3] | 40% | 80% × 40% | 0 | 59% |
| Full-fixed | Spring banning grazing [2] + Livestock-rearing control [3] | 50% | 80% × 40% | 0 | 66% |

[1] Banning grazing all year round means that the grassland resources cannot be used for grazing throughout the year to accelerate the recovery of grassland vegetation. [2] Spring banning grazing means that grazing is not allowed in the grasslands during spring. [3] Livestock-rearing control means lowering human intervention in grassland by limiting the number of livestock allowed to graze in a given area.

### 3.1.3. Spatiotemporal Utilization Patterns in Typical Steppe

The grazing utilization strategies and proper use factors in typical steppe are shown in Table 2. Non- and lightly degraded steppe have high vegetation coverage and vigorous vegetation growth. Grazing should be prohibited in spring, and livestock-rearing control is carried out in other seasons. When steppe suffers medium degradation, although vegetation cover is also relatively high, it is recommended to implement spring banning grazing and livestock-rearing control. Severely degraded steppe has low grass coverage and a large reduction in grass yield and banning grazing all year round must be carried out.

**Table 2.** Spatiotemporal utilization patterns in typical steppe.

| Degradation Conditions | Grazing Strategies | Proper Use Factors | | | |
|---|---|---|---|---|---|
| | | Summer | Autumn and Winter | Spring | Annual |
| Severe degradation | Banning grazing all year round [1] | 0 | 0 | 0 | 0 |
| Medium degradation | Spring banning grazing [2] + Livestock-rearing control [3] | 20% | 80% × 20% | 0 | 33% |
| Light degradation | Spring banning grazing [2] + Livestock-rearing control [3] | 40% | 80% × 30% | 0 | 54% |
| No degradation | Spring banning grazing [2] + Livestock-rearing control [3] | 50% | 80% × 40% | 0 | 66% |

[1] Banning grazing all year round means that the grassland resources cannot be used for grazing throughout the year to accelerate the recovery of grassland vegetation. [2] Spring banning grazing means that grazing is not allowed in the grasslands during spring. [3] Livestock-rearing control means lowering human intervention in grassland by limiting the number of livestock allowed to graze in a given area.

### 3.2. Available Grass Yield of Grassland

The grass yield was estimated for the 3 grazing seasons and the whole year using equation (1) based on the NPP prediction results. The accuracy of the model-estimated grass yield was validated using in situ measured data, and a RMSE of 47.47 g/m$^2$ and a MAE value of 37.27 g/m$^2$ were obtained, which showed that the estimated grass yield can be used to calculate available grass yield.

In Zhenglan Banner, total annual available grass yield was 625.76 kt, of which the available grass yield in summer was 416.78 kt, and the available grass yield in autumn and winter was 208.98 kt. The spatial distribution of available grass yield was similar in the grazing season and throughout the year (Figure 2). The areas with lower available grass yield were mainly clustered in the northwest and northeast, while the areas with higher available grass yield were distributed in the north and southeast. Through the regulation of spatiotemporal utilization pattern for grassland resources, the available grass yield offered by severely degraded steppe, shifting and semi-shifting sandy land was 0 g/m$^2$ throughout the year, and the available grass yield provided by all other grassland resource types was also 0 g/m$^2$ in spring. Full-fixed sandy land provided the highest available grass yield throughout the year and in autumn and winter, with values of 123.46 g/m$^2$ and 33.85 g/m$^2$, respectively; and in summer no degraded steppe supplied the highest available grass yield (90.00 g/m$^2$) (Table 3).

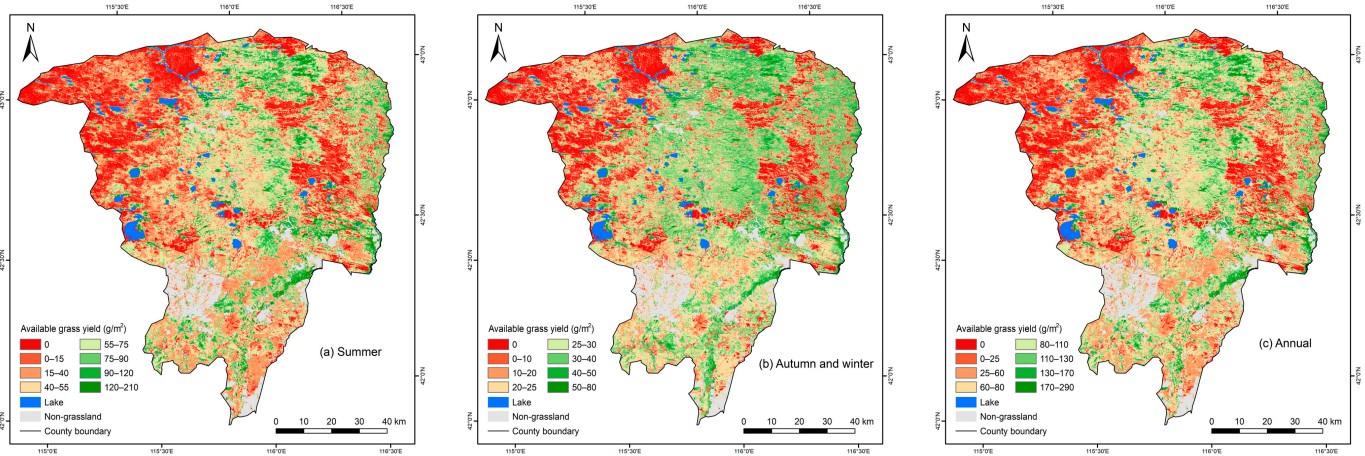

**Figure 2.** Spatial distribution of annual available grass yield and its distribution in different grazing seasons.

**Table 3.** Average available grass yield in different grassland types.

| Grassland Resource Conditions | Average Available Grass Yield (g/m$^2$) | | | |
|---|---|---|---|---|
| | Spring | Summer | Autumn and Winter | Annual |
| Full-fixed sandy land | 0 | 89.60 | 33.85 | 123.46 |
| Fixed sandy land | 0 | 61.19 | 33.83 | 95.03 |
| Semi-fixed sandy land | 0 | 38.96 | 24.86 | 63.82 |
| Semi-shifting sandy land | 0 | 0 | 0 | 0 |
| Shifting sandy land | 0 | 0 | 0 | 0 |
| No degraded steppe | 0 | 90.00 | 33.17 | 123.17 |
| Lightly degraded steppe | 0 | 66.03 | 26.87 | 92.90 |
| Medium degraded steppe | 0 | 32.71 | 22.80 | 55.52 |
| Severely degraded steppe | 0 | 0 | 0 | 0 |

### 3.3. Livestock Carrying Capacity for Grassland

Livestock carrying capacity for grassland varied in different time periods. In summer, grassland had the highest livestock carrying capacity, with a livestock number of 237.46 thousand cattle-units. In autumn and winter, the carrying capacity of grassland

was the lowest, with a total of 79.81 thousand cattle-units. This means that if the livestock number was maintained except for summer, large number of other forage and fodder must be supplemented. The spatial distribution of livestock carrying capacity in different grazing seasons can be seen in Figure 3.

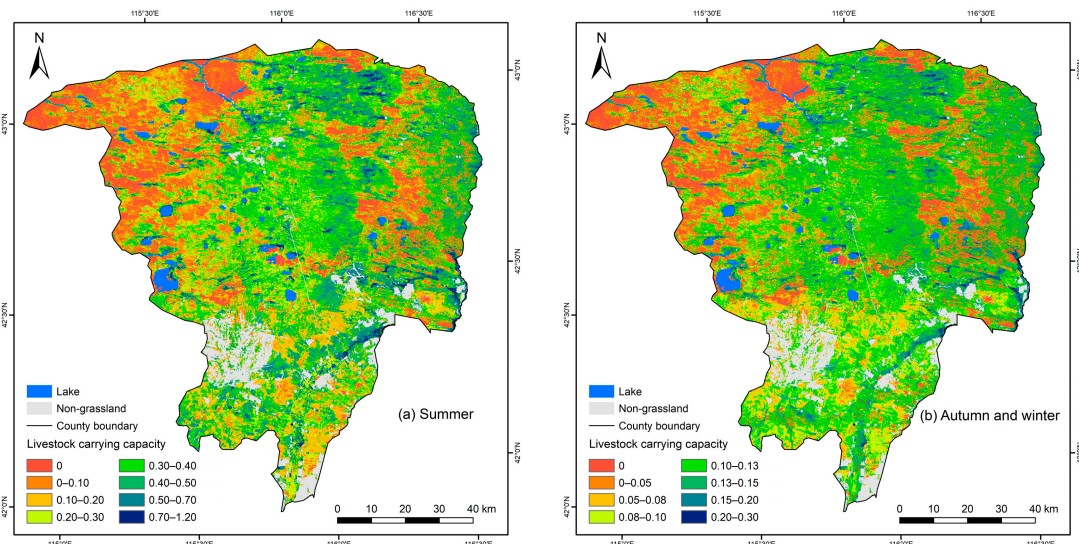

**Figure 3.** Assessed livestock carrying capacity (unit: cattle-unit/ha) throughout grazing season according to the carrying capacity model that considers available grass yield determined for different grassland resources conditions as in Figure 2.

At the Gacha-scale, in areas where grazing was allowed, the livestock density in summer ranged from 0.06 cattle-unit/ha in Wuritutala to 0.42 cattle-unit/ha in Caiyuan. The Wuyi breeding farm could carry the largest number of livestock at 18,847 cattle-units, more than double the number of livestock carried by Narisitu, which carried the second largest number of livestock. In autumn and winter, the Gacha with the highest and lowest stocking density was consistent with that in summer. The stocking density in autumn and winter ranged from 0.03 cattle-unit/ha to 0.13 cattle-unit/ha. Figure 4 demonstrates the gradual increasing trend of livestock carrying capacity radiating outward from the southeast corner. The northwestern Gachas had the lowest carrying capacity, with a carrying capacity of less than 0.2 cattle-unit/ha in summer, less than 0.06 cattle-unit/ha in autumn and winter.

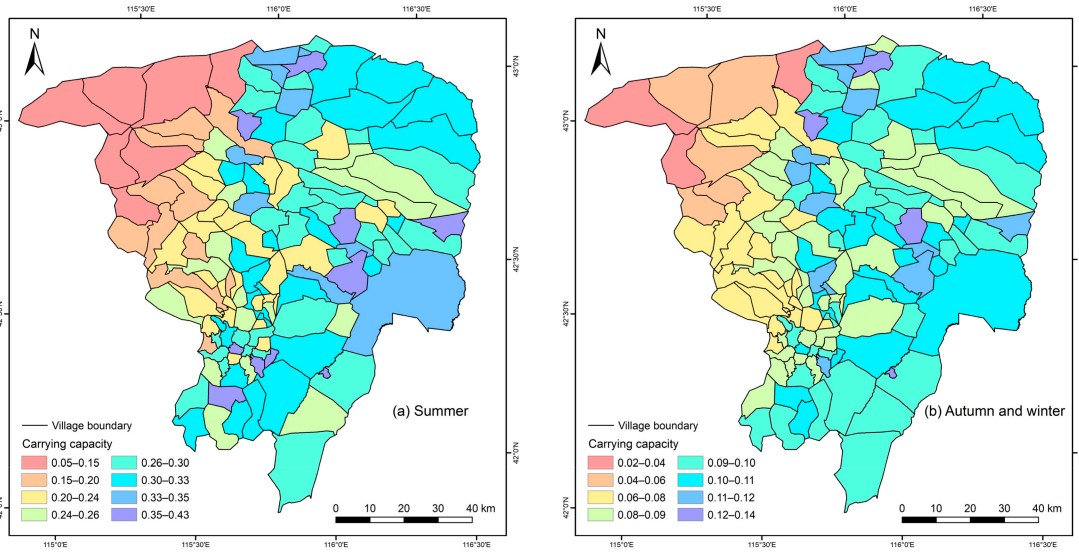

**Figure 4.** Spatial distribution of livestock carrying capacity (unit: cattle-unit/ha) in different grazing seasons at the Gacha-scale.

### 3.4. Current Status of Livestock Overload

To provide a more realistic picture of the current livestock carrying status, the livestock carrying capacity in summer was taken for further evaluation. Because livestock feed mainly comes from natural grassland in summer, supplementary feeding is necessary in other grazing seasons. The actual stocking rate of 109 Gachas in Zhenglan Banner totaled 283.32 thousand cattle-units in summer. The overload rate was 19.32% compared to carrying capacity in summer. At the Gacha-scale, the results of the livestock overloading rate assessment revealed that 71 Gachas exhibited livestock overload. Twenty-four Gachas had an overloading rate greater than 100%, 18 Gachas had a livestock overloading rate between −10% and 10%, where basic balance was maintained between grass and livestock. Additionally, 20 Gachas still had some grazing potential. The spatial distribution of actual stocking rate and livestock overloading rates demonstrates that the Gachas with higher stocking rate were mainly concentrated in the southwestern part (Figure 5).

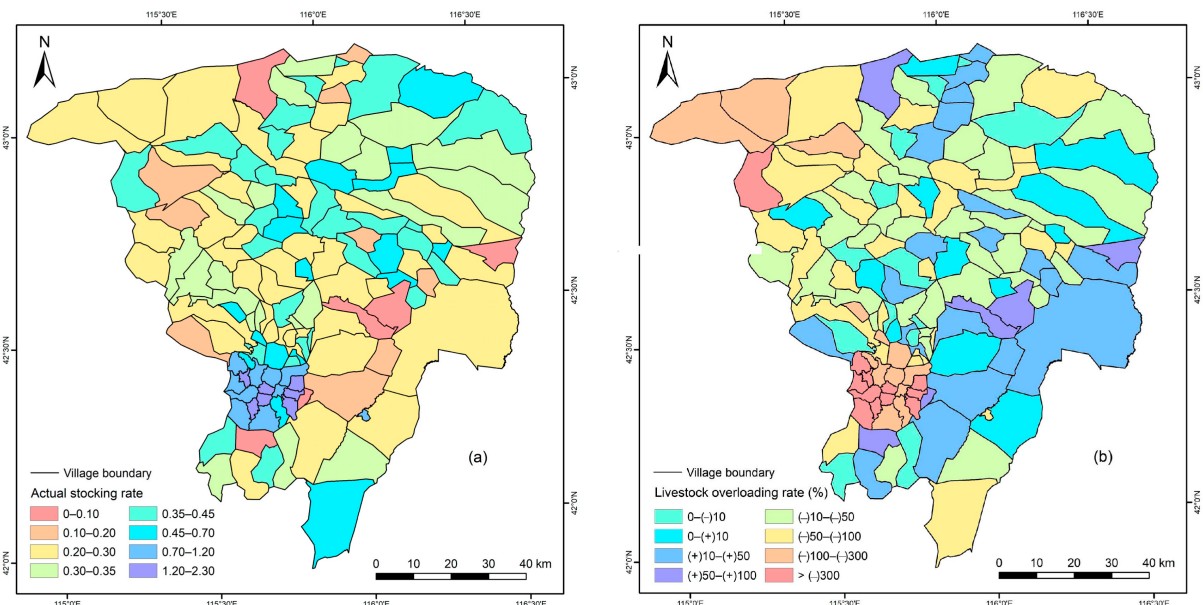

**Figure 5.** Spatial distribution of (**a**) actual stocking rate (unit: cattle-unit/ha) and (**b**) livestock overloading rate in summer at the Gacha-scale.

## 4. Discussion

### 4.1. Improvement of the NPP-Based Grass Yield Estimation Model

Zhenglan Banner is located in an arid and semi-arid region, with large seasonal variations in grass yield, accompanied by significant spatial heterogeneity [104,105]. The NPP-based grass yield estimation model in this study constructed by fusing high spatial-temporal resolution remotely sensed data and optimized CASA model could achieve more detailed spatial delineation and capture intra-annual dynamic. In addition, the stability and transferability of the estimation model are better than those of the widely used statistical models. On the other hand, the high spatial resolution of grass yield estimation results also contribute to improve the accuracy of livestock carrying capacity assessment at the Gacha-scale. Based on this, we conclude that the grass yield estimation model has the appeal of being implemented in drylands.

The accuracy of the grass yield estimation model depends mainly on the NPP estimation error. In the CASA model used to estimate NPP in this study, modelling of different maximum light use efficiencies according to different vegetation characteristics can minimize estimation errors. In the original CASA model, the maximum light use efficiency is set to 0.389 gC·MJ$^{-1}$ for all vegetation types in the world [106,107]. However, it was proved that this constant assignment is not appropriate in different vegetation types as

it may be affected by various influencing factors, such as local temperature, moisture, soil conditions, and plant individual development [108–110]. Many previous studies involving the study area estimated the maximum light use efficiency for grassland with reference to the results of a large-scale field survey, and with values ranging from 0.45 to 0.542 gC·MJ$^{-1}$ [74,110,111]. In this study, the maximum light energy use efficiency of grassland, which was re-optimized for the sparse vegetation cover and relatively complex surface conditions in the study area, was determined from our previous study [96,97] with a value of 0.518 gC·MJ$^{-1}$, which is similar to the results of published studies by other scholars.

The proportion of NPP allocated to above-ground growth has received far less attention [112,113] and is likely to be an uncertain parameter in the estimation of grass yield. This study directly used the default proportional factor in semi-arid grassland reported by the Intergovernmental Panel on Climate Change (IPCC) [94] as a model parameter for the entire study area. While at the community level, the NPP allocation proportions still differs across grassland types [114–116]. Future research will need to determine the proportion of NPP allocated to the above-ground component of different grassland vegetation types by obtaining measured data of below- and above-ground growth as a way to improve estimation accuracy and provide support for long-term grass yield estimation.

When this model is desired to be implemented in different places, its good performance in application in the grasslands of Zhenglan Banner does not necessarily mean that it will function with the same accuracy in grassland biomes elsewhere. This is because in the process of grass yield estimation in typical steppe, although the vegetation homogenization is relatively high, there are still differences in vegetation types. Since the typical steppe area in this study is small, the vegetation types are less different and all belong to typical steppe types, and the grass yield estimation model also takes into account the influence of non-vegetation factors such as topography and climate, so it does not have much influence on the estimation accuracy. Given this uncertainty, we recommend that when implementing this estimation model in other situations, efforts should be made to validate its grass yield estimation results based on observations in the field.

### 4.2. Performance of the Grassland Resource Utilization Patterns

In Zhenglan Banner, grassland resources are mainly located in sandy land, followed by typical steppe (Figure 1c). According to the distribution characteristics of grassland in Zhenglan Banner, the key factors for sustainable utilization of grassland resources are the degree of sand mobility and the degradation condition of typical steppe. In this study, the grazing strategies and proper use factors were developed based on the two factors—namely, the degree of sand mobility and the degradation condition of typical steppe, with the aim of coping with the spatial heterogeneity of grassland resources and achieving the sustainable utilization of grassland resources. On this basis, the intensity of grassland utilization was developed by dividing grazing seasons with the purpose of preventing seasonal overload of grassland resources. The maximum value of the proper use factors in the pattern were proved to be reasonable by comparison with the grassland proper use factors used in published livestock carrying capacity studies in the study area and the region covering the study area [84,117–119].

There is still room for further refinement of the pattern, for example, by considering the species composition, nutrient quality, and soil condition of grassland resources [29]. The spatial and temporal variation in species composition of grassland resources is an essential step in evaluating grassland health and developing grassland management strategies and can be achieved by high spatial resolution image classification [120]. The nutritional quality of grassland directly affects the growth of livestock and the quality of livestock products. The use of hyperspectral remote sensing technology may deeply depict the dynamic changes in the nutritional quality of natural grassland [121]. Soil condition is a key driver for grassland growth and quality [122]. The availability of global soil maps that provide such information, such as SoilGrids [123], will likely include the effects of soil

texture or K-factor or soil erodibility, which have recently been modeled using soil texture data from SoilGrids [124]. These factors will be considered when developing grassland utilization patterns in future work. Meanwhile, how to weigh the synergistic relationship of each impact factor and achieve the integrated use of factors is also a key we are prepared to address subsequently.

When the pattern was applied to the summer grassland of Zhenglan Banner, our carrying capacity assessment resulted in an average stocking density of 0.26 cattle-unit/ha. This figure is slightly below the stocking density for the summer grassland prescribed by the Zhenglan Banner government of 0.3 cattle-unit/ha [125]. The difference is understandable because our assessment focused on achieving ecological restoration. Zhang et al. [126] reviewed the typical grassland stocking density proposed in Inner Mongolia ranging from 0.25 to 0.48 cattle-unit/ha, which is consistent with the stocking density in this paper. The assessment results of this study provide information on geographic and seasonal variations in carrying capacity, which can support stakeholder decisions, such as implementing more sustainable stocking rates, determining reasonable number of livestock for sale, and providing some reference to the amount of fodder purchased. In addition, the application of high spatiotemporal remote sensing technology can ensure that livestock carrying capacity estimations at the spatial scale of the Gacha is more refined, which facilitates the implementation of grassland utilization strategies.

The grassland resource utilization patterns of this study were constructed for the animal husbandry dominated by grazing, which can be better applied to grazing livestock production system with a long production cycle (winter and spring can be supplemented with some concentrates) and seasonal rest grazing production system widely implemented in grassland pastoral areas. Hence, this method does not apply to livestock production systems based on grain concentrates and straw fodder in agricultural areas and to short-term fattening production systems. On the other hand, these two livestock production systems have a little direct impact on grassland, so there is no need for general grass–livestock balance management [127]. When we conducted a reasonableness analysis of the Gachas with overload rates greater than 150% in Zhenglan Banner, it was found that (Figure 5b), except for Bayanhanggai Gacha and Bayannaoer Gacha, all of the Gachas were located in agricultural areas where livestock are raised on grain concentrates and straw fodder. These results are not indicative because the estimated carrying capacity was not suitable for use in this area. Exclusion of the Gachas where carrying capacity estimates are not applicable, 51 Gachas exhibited livestock overload.

Zhenglan Banner is a typical pastoral area. More than 70% of the total population are herdsmen whose income mainly depends on traditional grassland animal husbandry [128]. If the number of livestock is controlled strictly according to the grassland resource utilization pattern of this study, it will certainly affect the income of herders, thus making the implementation of the grassland utilization model more difficult. Therefore, while implementing the regulation of a grassland resource utilization pattern, the local government should promote the transformation of traditional grassland animal husbandry to modern grassland ecological animal husbandry with large-scale breeding and intensive operation. By cultivating small areas of high-yielding and high-efficiency artificial grassland, the high-quality forage needed for livestock development can be met [5,9,129]. By exploiting the ecological tourism value and developing special forest, sand, and grass industries, the income of local herdsmen is improved. In this case, the regulation of grassland resource utilization pattern can be implemented in a sustainable manner.

## 5. Conclusions

To achieve sustainable use of grassland resources, we developed the grassland resource utilization patterns for three grazing seasons based on the degree of sand mobility and typical steppe degradation and completed the estimation of grassland carrying capacity. Through the regulation of spatiotemporal utilization pattern, the annual available grass yield in Zhenglan Banner was estimated to be 625.76 kt, with 0 kt of available grass

yield in spring, 416.78 kt in summer, and 208.98 kt in autumn and winter. The grassland carrying capacity in summer was 237.46 thousand cattle-units. The grassland carrying capacity in autumn and winter was 79.81 thousand cattle-units. The grassland resources of Zhenglan Banner are being overgrazed, with an overloading rate of 19.32%. Fifty-one Gachas exhibited livestock overload. It is advisable to reduce the current stocking density.

The spatiotemporal utilization patterns for grassland resources proposed in this paper have the potential to be replicated in other regions. The utilization pattern is not only able to cope with the high spatial and temporal heterogeneity of grasslands in drylands but also contributes to the ecological restoration of grasslands. Grassland degradation is a widespread problem in northern grasslands in China, and we therefore suggest using this pattern to provide technical support for the entire northern dryland region of China. Nevertheless, there is still room for further improvement in the accuracy of grassland carrying capacity estimation. In our future work, we will improve the accuracy of grass yield estimation and perfect spatiotemporal utilization patterns for grassland resources.

**Author Contributions:** Conceptualization, Z.G., P.Q., and Z.L.; methodology, Z.G. and B.S.; software, Y.L. and P.Q.; validation, T.G., Z.Y., and P.Q.; formal analysis, B.S. and Z.G.; investigation, P.Q., B.S., Z.G., Y.L., Z.Y., and T.G.; resources, Z.G. and B.S.; data curation, Y.L. and Z.Y.; writing—original draft preparation, P.Q.; writing—review and editing, B.S., P.Q., and Z.G.; visualization, P.Q.; supervision, Z.G. and Z.L. All authors have read and agreed to the published version of the manuscript.

**Funding:** This research was supported by "the Fundamental Research Funds for the Central Non-profit Research Institution of CAF", grant number CAFYBB2020ZB001, Inner Mongolia Autonomous Region Financial Special Project.

**Institutional Review Board Statement:** Not applicable.

**Informed Consent Statement:** Not applicable.

**Data Availability Statement:** Data can be provided upon request from the corresponding author.

**Acknowledgments:** We would like to thank the government of Zhenglan Banner for its full support and cooperation in the field investigations and basic data acquisition.

**Conflicts of Interest:** The authors declare no conflict of interest.

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
