# Peer review of "Estimation of Grassland Carrying Capacity by Applying High Spatiotemporal Remote Sensing Techniques in Zhenglan Banner, Inner Mongolia, China"

_sustainability, doi:10.3390/su13063123_

Round 1

Reviewer 1 Report

The authors presented a study about the capacity of grasslands and their current use to highlight the overuse of natural resources. The topic is interesting, novel, and fits with the journal scope. In general terms, their study has a high relevance; nonetheless, there are some issues to be solved in the manuscript. There are some small issues related to the forma and to the English, which must be solved in order to enhance the quality of the paper. Following, I include my recommended modifications: 

The paper must be proofread, there are several sentences extremely long which difficulties the reading of the paper.  

The abstract is too long. Authors must reduce the abstract to 200 – 250 words ensuring that they highlight their results.  

In keywords authors should avoid using the same word that in the title. 

In the introduction, the authors include a reference by adding the webpage, this should be added as a regular reference “[1]”. See line 41-42. 

I suggest adding the aim of the paper as well as the objectives in an independent paragraph. Moreover, the structure of the paper can be included in a new paragraph after the objectives. 

In Equations, authors have to include the units of the variables. 

Authors must check the Figures caption, there is an error which should be corrected (there are two Figures named Figure 1).  

Table 3 should be modified to include the available grass yield and its units in the table not only in the title. 

Check the sentence in line 606, data is missing for winter. In line 608, units are missing. Please check similar cases. 

In the conclusion section, authors have to include their future work linked to their findings. 

Reviewer 2 Report

The manuscript entitled „Estimation of grassland carrying capacity by applying high spatiotemporal remote sensing techniques in Zhenglan Banner, Inner Mongolia, China” by Pengyao Qin et al. investigates an important and actual topic related to grassland science. The formulation of grassland utilization strategy and livestock carrying capacity assessment were performed following a novel approach - high spatial and temporal resolution technique which is very good. The authors reached interesting results relevant to the scientific discoveries in terms of their applications. From my point of view this research fits very well with Sustainability journal profile.

I recommend that authors should review their manuscript in term of English language and formatting:

  1. Line 42- the link http://www.fao.org should be numbered and introduced into references section (is totally missing from references)
  2. Line 44-45- “soil and water conservation, water conservation”- authors should delete water conservation since appears twice in the same context
  3. Line 425, 440, 457, 473- wrong figure numbering- starting with Figure no. 2 all the figures should be re/numbered
  4. Line 594 – reformulate “The main conclusions of the study as follows”.
  5. References:
  • Line 138- The authors cited Yu et al.[78]- but the reference no. 78 is Long et al. instead yu et al appears listed in 13 and 63 position in Reference section.
  • Line 684- “Rangeland …: 2009.”- is it ok like this?
